# Tri-Response Police, Ambulance, Mental Health Crisis Models in Reducing Involuntary Detentions of Mentally Ill People: A Systematic Review

**Julia Heffernan** [1,*]**, Ewan McDonald** [1]**, Elizabeth Hughes** [2] **and Richard Gray** [1]

1   School of Nursing and Midwifery, La Trobe University, Melbourne, VIC 3086, Australia
2   School of Health and Social Care, Edinburgh Napier University, Edinburgh EH11 4BN, UK
*   Correspondence: j.heffernan@latrobe.edu.au

**Abstract:** Police, ambulance, and mental health tri-response services are a relatively new model of responding to people experiencing mental health crisis in the community, though limited evidence exists examining their efficacy. Reducing unnecessary involuntary detentions and emergency department presentations is believed to be a benefit of this model. A systematic review was performed to review the evidence base around the relationship between the police, ambulance, mental health tri-response models in reducing involuntary detentions of people experiencing mental health crisis. We searched key health databases for clinical studies and grey literature as per a previously published protocol. Two researchers completed title and abstract screening and full text screening. Our search identified 239 citations. No studies or grey literature met the inclusion criteria. We report an empty review. It is recommended that further investigation of the tri-response mental health crisis model be undertaken to determine its effectiveness and value as a health and emergency service initiative.

**Keywords:** systematic review; police; ambulance; mental illness; involuntary detention; section

## 1. Background

Tri-response mental health models use a police officer, ambulance paramedic and mental health clinician/nurse to jointly attend people experiencing a mental health crisis in the community. It is theorized that inserting mental health expertise through the provision of rapid mental health assessment, can improve patient outcomes by reducing the use of restrictive practices such as involuntary detention to enforce compulsory hospital assessment.

The objective of this review was to synthesise the available evidence regarding the effects of tri-response police, ambulance, mental health crisis models in diverting patients from hospital and reducing unnecessary involuntary detentions.

## 2. Rationale

Involuntary detention is a common mechanism used to compel people who appear acutely mentally ill for a mandatory psychiatric assessment or period of observation in hospital [1–6]. It is a controversial power provided in mental health legislation, generally to doctors, mental health workers, police officers and in some areas, paramedics. Involuntarily detaining a person brings with it additional powers which allow force to be used upon the person to complete that detention as included in mental health legislation [4–6]. Such powers may include forcing entry into the persons property, searching their person and property, using physical force and restraint, and the use of chemical sedation [1–5].

In developed countries, mental health legislation is similar in content and allows for involuntary detention, transport, assessment, treatment and hospitalization for those who do not have decision making capacity due to mental illness [6]. However, for police and paramedics with mental health legislative powers, the use of clinical judgement in detaining

a patient is not generally required under legislation, and this power can be invoked using a lay person's judgement that the person is mentally ill [7].

Mental health patients and carers report that involuntary detention is a traumatic, humiliating and often frightening experience, particularly when involving police or law enforcement agencies, which negatively impacts their overall mental wellbeing [1,8–13]. It is consistently demonstrated that involuntary detention invokes loss of perceived independence, worsening of paranoid beliefs, terror and distress, re-traumatisation and powerlessness, particularly for those who experienced restrictive practices such as restraint and forcible giving of medications [1,8,10,12].

Police will often invoke involuntary detention in the absence of other mechanisms to ensure prompt assessment of a person experiencing mental illness [2,14,15]. Similarly, ambulance paramedics frequently respond to mental health crisis, often co-responding with police [16,17]. High rates of involuntary detentions have significant resourcing impacts on emergency services and hospital emergency departments who are required to undertake the assessments. Such as higher numbers of patients in emergency departments (EDs), the need for greater supervision, prevention of absconding, and pressures related to involuntary detention assessment times [18–22].

Responding to mental health crisis presentations has become considerably more frequent for paramedics, yet there has been little focus on providing appropriate mental health training [16,17,23]. A scoping review that included fourteen peer-reviewed articles examined paramedic management of mental health presentations [23]. The authors noted important gaps in education and training, organisational and operational factors and clinical decision making, relating to mental health presentations [23]. A prospective cohort study in Australia examined the rate of mental health patient admission following involuntary detention by paramedics to the Royal Prince Alfred Hospital Emergency Department [24]. The study found that 27% of patients involuntarily detained by paramedics, went on to have a hospitalization as compared to 60% when invoked by medical practitioners and other accredited persons [24]. The authors determined that paramedic involuntary detention was a poor predictor of the need for hospitalization and also noted the extended period of time that detained patients spend in the ED, contributing to access blocks and overcrowding [24].

Furthermore, a retrospective observational study of involuntary detentions by police in Australia determined that 67% of patients involuntarily detained by police, did not go onto be hospitalised, were deemed as not requiring immediate treatment or care, and were subsequently discharged [7]. The authors concluded that further exploration of less restrictive practices such as involuntary detention was warranted, particularly for people expressing thoughts of self-harm who made up the largest cohort of police involuntary detentions [7].

There is evidence from observational research of increasing rates of involuntary detentions over time [15,18–21,25]. For example, an analysis of involuntary detention rates in the United States reported that the rate of involuntary detentions in 22 states was increased year on year by 13% between 2012 to 2016 [19]. In Australia, involuntary detention rates were examined using a retrospective examination of emergency examination orders. The authors determined that the overall rate of involuntary detentions by police and paramedics had risen by 262% since 2002, with the proportion by paramedics increasing from 14.5% in 2004 to 38% in 2010, and police making up two-thirds of all involuntary detentions [22]. Furthermore, the rate of patients involuntarily detained by police and paramedics who were deemed as requiring hospital admission was less than half that of those detained [22].

A systematic review, meta-analysis and narrative synthesis of clinical and social factors associated with increased risk of involuntary psychiatric admission, included 77 studies which demonstrated an association between police involvement in admission and involuntary care, particularly in those presenting with psychotic illnesses [26].

As a result of increasing rates of involuntary detention; health services, police and ambulance services have observed the need to provide mental health expertise directly

into emergency service presentations. A trained mental health clinician can provide expert assessment in the field, negating the need to invoke involuntary detention or transporting the person to hospital for assessment [3,27]. A number of models are being trialed across the world including the addition of mental health workers into police or ambulance call centres, co-response mobile crisis services which may team a mental health worker with a paramedic or police officer, or the tri-response model which incorporates all three agencies [27].

The tri-response police, ambulance, mental health crisis model (the tri-response model) is operating in several Australian States and Territories and teams a mental health clinician, police officer and ambulance paramedic together in a first responder vehicle to attend mental health crisis in the community. Rather than sending a police or ambulance unit to attend a patient in mental health crisis, the tri-response model is sent to provide mental health expertise including mental state assessment, treatment and referral. It is one of several models being trialed to meet this need, yet it requires further exploration to assess its efficacy in reducing involuntary detentions. Other models include co-response models such as pairing a mental health clinician with a police officer or paramedic as a mobile crisis service or including them into police or ambulance communications centres.

A systematic review of police mental health co-responder models was undertaken in 2018 to identify and describe the different models, identify the types of service users who came in contact with the models and to evaluate their effectiveness [27]. The authors included 26 papers into the review and concluded that the co-responder police mental health models may reduce rates of involuntary detention of mentally ill people [27]. However, they reported that further research was required, and at the time of writing, no review has been conducted to evaluate the tri-service model [27]. This finding was further supported by a qualitative study of the experiences of Chicago police officers responding to mental health incidents, many of which described a "never ending cycle of hospital transport" and proposed that mental health nurses and co-responder models were well placed to support police officers in the field and to provide alternatives to emergency department transport [28].

It could be argued that a tri-response model comprising of a mental health clinician, a police officer, and a paramedic is an expensive model when compared with a co-response model. Furthermore, if the same outcomes of a tri-response model can be achieved using a co-response model, is the model justified? In many areas such as the United Kingdom, ambulance paramedics do not have powers under mental health legislation, so co-response police and mental health nurse are more prevalent.

## 3. Objectives

The specific objectives of the review were to synthesise the available evidence regarding the effects of the tri-response police, ambulance, mental health crisis model in diverting patients from hospital and reducing unnecessary involuntary detention.

1.  To identify the evidence base around the relationship of the tri-response police, ambulance, mental health crisis model in reducing involuntary detentions of people experiencing mental health crisis.
2.  To compare the rate of involuntary detentions by tri-response police, ambulance, mental health crisis model with rates of involuntary detentions made by police and/or ambulance paramedics.
3.  To compare tri-response police, ambulance, mental health crisis model involuntary detentions which result in hospitalisation, with those made by police and/or ambulance paramedics.

## 4. Materials and Methods

### 4.1. Protocol and Registrations

We report the method and outcomes of a systematic review. A detailed protocol for this review was published previously [29]. This review followed the Preferred Reporting Items for Systematic Reviews and Meta-Analyses (PRISMA) reporting guidelines [30].

The protocol was registered with OSF in May 2021 prior to the search being undertaken.

*4.2. Eligibility Criteria*

Our inclusion criteria were:
- Clinical studies of any design
- Grey literature including service evaluations, relevant theses or dissertations, research or committee reports, reference list review, citation searching and internet searching.
- The manuscript was written in English
- Involving patients of any age
- Involving the police, ambulance, clinician tri-response model
- Participants who experienced an acute mental health crisis which has precipitated an emergency response or the tri-response police, ambulance, mental health crisis model response.
- Involving manuscripts without date restriction
- There were no patient demographic restrictions
- We included any comparator in the review however the primary comparator was a standard police or ambulance emergency service response.

*4.3. Exclusion Criteria*

Our exclusion criteria were:
- Qualitative research

*4.4. Information Sources*

Searches were undertaken in October and November 2021 of the following databases: CINAHL, MEDLINE, PsychInfo, PsychArticles, Open Grey, Proquest and Google. We were unable to search the policing databases identified in Part One as they were not accessible to the author.

*4.5. Search Strategy*

Our search strategy was developed in collaboration with the Canberra Health Services Library Consultation Service and was developed in MEDLINE using medical subject headings (MeSH terms) and keywords which were adapted for the other databases. The full search strategy is available in the protocol [29].

*4.6. Study Selection*

Relevant studies and documents from databases were imported into EndNote X9 software by the Canberra Hospital Library consultant who removed duplicates. Studies and documents were then uploaded to the Covidence website (https://www.covidence.org accessed on 20 November 2021) and rechecked for duplicates. Covidence is a web-based software platform for systematic reviews including citation screening, review of full text articles, risk of bias assessment, extraction of study characteristics and outcomes, and exportation of data. Two reviewers from the research team completed title and abstract screening against the review inclusion criteria. Full-text review against the inclusion criteria was further undertaken by two reviewers of the research team.

*4.7. Data Selection Process*

Data extraction was to be completed in a Microsoft Excel file by two researchers.

*4.8. Data Items*

The following data items were extracted from the included studies and reports: author, country, study design, measure of involuntary detention, tri-response police, ambulance, mental health crisis model case numbers, tri-response police, ambulance, mental health crisis model involuntary detentions, emergency service case numbers, emergency service involuntary detentions, sample size.

*4.9. Risk of Bias in Individual Studies*

Included studies were to undergo quality appraisal using the Effective Public Health Practice Project Quality Assessment Tool (EPHPP) (https://merst.ca/wp-content/uploads/2018/02/quality-assessment-tool_2010.pdf). The EPHPP was developed in Canada by the Effective Public Health Practice Project and is an effective tool for evaluating a number of different study designs including Randomised Controlled Trials (RCTs), before and after intervention studies and case-control studies [31]. As the single piece of grey literature included in the review was not a clinical study, it was unable to be appraised using the EPHPP.

*4.10. Synthesis of Results*

We planned to undertake a narrative synthesis of results to focus upon the intervention implementation and effect, and group into themes.

**5. Results**

Figure 1 shows the flow of papers through the review process using the PRISMA Flow Diagram Policing databases listed in protocol were unable to be accessed for the literature search due to access limitations. This was the only change to the research methodology subsequent to publishing.

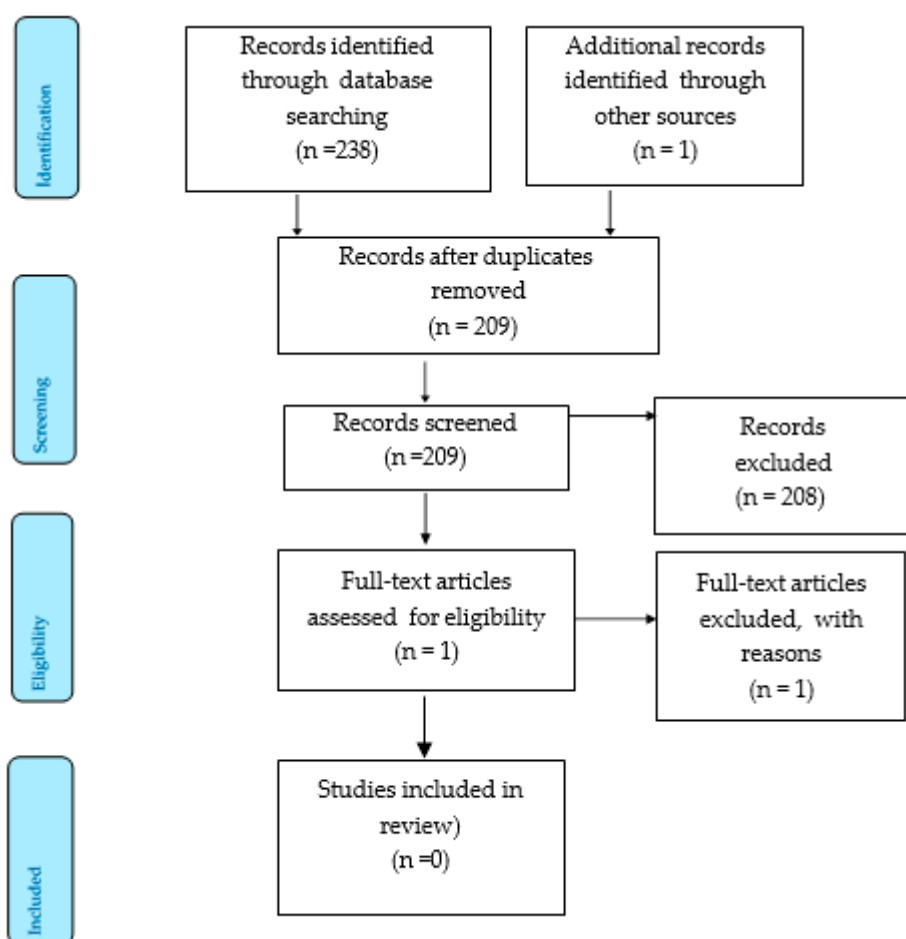

**Figure 1.** Preferred Reporting Items for Systematic Review and Meta-Analysis 2015.

Our initial search identified 239 citations, of which 30 duplicates were removed. A total of 208 citations were excluded during the title and abstract screening. The full text of one grey literature report was reviewed and excluded at full text screening. The primary

reason for exclusion of studies and reports was due to them examining co-response models, rather than a tri-response model. We provide a summary of the excluded full text paper:

In 2012, the Department of Health in Victoria, Australia, commissioned Allen Consulting Group to undertake an evaluation of their Police, Ambulance and Clinical Early Response (PACER) trial which operated from 2007 to 2011 [32]. This model, however, was a police and mental health mobile co-response unit operating across several areas in Victoria. Small discussion groups, interviews and stakeholder meetings with police, mental health services, ambulance services and mental health consumers, were conducted as part of the service evaluation.

The objectives of the evaluation were to evaluate the effectiveness and efficacy of the PACER model over a 16-month study period, in managing and resolving mental health crisis in the community compared with mental health, police and ambulance service response. The group also completed a cost effectiveness analysis of the PACER model, compared with standard emergency services responses, and to identify enablers and challenges in implementation (p. 10).

The Victorian PACER model was piloted following the Victorian Health Reform Strategy 2009–2019: Because mental health matters (Department of Human Services 2009) which demonstrated a need for more integrated approaches to mental health crisis and promotion of least restrictive practices and greater capacity for community treatment options.

The evaluation framework developed for the PACER evaluation was as follows:

- What impact has PACER had on facilitating a coordinated response that provides more timely access to mental health patient care?
- Does PACER provide a more streamlined and quality approach to mental health crisis in the community through improved access to mental health advice and agency client histories?
- Are there fewer adverse events in the community arising from management of emergency mental health crises?
- Is there a reduced demand on agency resources in responding to emergency mental health crisis?
- Is there a reduction in referrals of people experiencing mental health crisis to emergency departments and an increase in referrals to other non-emergency services or direct admission to psychiatric inpatient facilities?

The methodology for evaluating the PACER project was a mix of qualitative and quantitative approaches applied to a comparator analysis of the PACER project to usual service provision in a comparator site. Data was sourced from service utilisation datasets regarding information about activity, outputs and outcomes related to emergency services involved in responding to mental health crises, such as PACER data activity sheets, patient transport forms, emergency department data, and information from the Victorian Ambulance Clinical Information System (VACIS).

During the evaluation period, PACER attended 783 requests for assistance, averaging 2 cases per 8 h shift. The primary response unit (police or ambulance) was cleared by PACER in 53 percent of cases. 37 percent of cases were involuntarily detained, and transportation was not required in 64 percent of cases [32]. Transport to hospital was reduced from 82 percent in the comparator group to 52 percent in the PACER group [32]. Difficulties in matching points of alignment across datasets affected the robustness of the cost effectiveness analysis.

The following key findings of the evaluation were:

- PACER provided more timely access to mental health crisis assessment, reduced the average time of assessment from 3 h to one hour, when compared to emergency services comparator.
- On average, PACER was able to release first responding police units faster and was the first responding unit approximately one third of the time, allowing police services to meet other demands.

- Where patient transport was required, ambulance services were utilised more often with PACER, for a person experiencing mental health crisis who required hospital transport. This is a preferred method of transport and is consistent with a least restrictive approach and preferable to police transport.
- Within the PACER model, there were fewer referrals to hospital emergency departments than in the comparator site, reflecting greater capacity for community treatment options.

Overall, when compared with standard police and ambulance response, the PACER model was more effective in timeliness to assessment, earlier clearance of first responders, achieved a more integrated management approach to mental health crisis, improved use of agency resources such as ambulance transport, and fewer transports to the hospital emergency departments [32]. Furthermore, based upon cost effectiveness analysis, the PACER model was less costly that standard service provision [32].

## 6. Discussion

The systematic review aimed to identify and synthesise the evidence for the association between the tri-response mental health crisis model and involuntary detentions. No clinical studies or grey literature was identified that met the inclusion criteria. Consequently, we report this as an empty systematic review. There is an important gap in the literature regarding the efficacy of tri-response mental health crisis models and supports and justifies the need for further research into this topic. Tri-response police, ambulance, mental health crisis models are operating in Australia, the United Kingdom and Canada with no evidence of their efficacy, safety or value.

There is some debate as to the value of empty systematic reviews. An empty review is defined as a systematic review that finds no eligible studies [33]. The authors surmised that empty reviews may appear to: "(1) offer no conclusions, (2) offer conclusions based on referenced excluded studies, (3) offer conclusions based on other evidence, or (4) offer conclusion not based upon evidence" [33] (p. 1). The authors went on to estimate that as many as 1 in 10 Cochrane reviews, were empty and most are denied publishing rights. However, Gray [34] argues that empty reviews do have value and should be published if they have been conducted with strong methodology and rigor. Empty reviews contribute to the body of knowledge on a topic by highlighting a gap in the literature and an opportunity for further research [34]. Furthermore, an editorial published in the Joanna Briggs Institute of Systematic Reviews and Implementation Reports argued that whilst empty systematic reviews were unable to provide recommendations for practice related to the initial review question, they still have important implications [35]. These can be described as providing direction for research to fill a gap in knowledge, with recommendations to guide the types of research designs needed in the future, recommendations about eligibility criteria for sample selection as specific interventions, comparators and outcome measures for future research into the topic [35].

## 7. Review Limitations

Our review did include some limitations which are important to consider. Given that our review did not generate enough literature to undertake a synthesis, it could be argued that a scoping review would have been more appropriate than a systematic review. Scoping reviews have become increasingly popular over the past decade, particularly in the disciplines of health and social sciences [36–38].

Scoping reviews may be useful for developing research questions and objectives, protocol development, planning the research approach and summarizing evidence and is also often used as a precursor to a systematic review [36]. Scoping reviews can assist in planning an initial research project but are not suitable for all research questions [36,37].

We consider that a systematic review was appropriate as the intention is to produce evidence that can inform policy decisions about feasibility, appropriateness and effectiveness of a particular treatment or therapy [36–38]. Furthermore, systematic reviews are

more widely used as they follow a structured and pre-defined process using a systematic method with a view to minimise bias and producing robust and reliable findings [37]. As their name suggests, scoping reviews may be a valuable tool to scope the availability and volume of literature, particularly for evidence that is emerging and when it is unclear of more specific questions can be addressed in a more structured process [36,37]. According to the literature, a systematic review should be used when the researcher is seeking to answer a clinically meaningful question which addresses the feasibility, effectiveness or appropriateness of a treatment, intervention, or practice [36,37]. We argue that we have developed a well-considered research question in collaboration with a systematic review consultant, which seeks to evaluate the effectiveness of the tri-response police, ambulance, mental health crisis model in reducing involuntary detentions, and therefore, the selection of a systematic review rather than a scoping review was appropriate.

A further limitation of our review was the inclusion of grey literature. Grey literature is a broad term that includes a large range of documents such as unpublished studies, government reports, and dissertations, that are not generally controlled by commercial publishing organisation. This means that grey literature can be difficult to search and retrieve for evidence synthesis [39,40].

## 8. Conclusions

There are no clinical studies or grey literature to evaluate the effectiveness of police, ambulance, mental health tri-response models in reducing involuntary detentions of people experiencing mental health crisis. This represents a gap in the evidence base which would benefit from further research.

**Author Contributions:** J.H. conceptualized, prepared, and wrote the review and original draft. R.G., E.M. and E.H. supervised the review and edited the manuscript. All authors independently reviewed the search strategy. All authors have read and agreed to the published version of the manuscript.

**Funding:** This research received no external funding.

**Institutional Review Board Statement:** Not applicable.

**Informed Consent Statement:** Not applicable.

**Acknowledgments:** The authors acknowledge the Canberra Hospital Library Consultancy Service for undertaking the literature search.

**Conflicts of Interest:** The authors declare no conflict of interest.

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
