# Peer review of "Tri-Response Police, Ambulance, Mental Health Crisis Models in Reducing Involuntary Detentions of Mentally Ill People: A Systematic Review"

_nursrep, doi:10.3390/nursrep12040096_

Round 1
Reviewer 1 Report (New Reviewer)
The manuscrit consist in a systematic review entailing the tri-response model implanted in Australia to attend mental healht crisis. This is an empty reiview. For that no significant results could be showed, only show a field of interesting future research.
The paper was well-writted and justified, and the methodology is robust since it has been published previously.
I’ll really think that autors could be designed the review as with a more general objective including the response maded by several models (including the tri-response). Possible in this case results will be diferent.
The introduction and discussion should be improved according studies carried out wiht other models of resposne to attend mental health crisis.
Author Response
Please see our responses to your review with thanks

Reviewer 2 Report (Previous Reviewer 3)
detentions of mentally ill people.
The paper refers to an exciting and relevant research topic with significant theoretical and practical implications. Although it is an empty systematic review, I enjoyed reading the paper and I consider the authors were able to justify its results presenting a very clear, objective and methodologically accurate manuscript.
Although its strengths, I will present some small constructive comments and suggestions which may help the authors to improve their work.
- I consider the abstract and the 1. Background sections need further development and clarification.
- The background section doesn’t bring any added value and only confuses the reader regarding the difference between PACER and Tri-response models. The abstract should be more clear in terms of novelty, goal, main results and practical implications.
- In the beginning of the paper, it is not clear to me the relationship / difference between Police, Ambulance, Clinician, Early Response (PACER) mobile crisis service and similar tri-response models. This clarification should be present both in the abstract and in the background section.
- The reference list is not according to the journal guidelines.
I hope these comments and suggestions help the authors to improve the quality and readability of their manuscript.
Good luck!
Author Response
Please see our response to your review with thanks

Round 2
Reviewer 2 Report (Previous Reviewer 3)
Thank you for the opportunity to read, once again, this paper. After careful analysis, I believe that the manuscript is ready for publication. From my point of view, the authors were able to answer to reviewers' questions and suggestions.
Thank you for the opportunity to contribute.
I wish all the best for the authors.
This manuscript is a resubmission of an earlier submission. The following is a list of the peer review reports and author responses from that submission.
Round 1
Reviewer 1 Report
Although the results obtained are not conclusive, the article is still interesting to read. However, It would be interesting if the authors proposed avenues of research to measure the effectiveness of the tri-model compared to the duo team model. I therefore suggest that the authors improve their discussion and conclusion accordingly. As it stands, the paper contributes very little to the advancement of knowledge. I wonder about the relevance of publishing these results. Rather, the findings should be incorporated into a comparative study to support their relevance.
Reviewer 2 Report
Dear Sir/Mam
Please find bellow the requested review regarding the manuscript. The article contains a lot of useful information on the issue. The topic is very interesting and use of sources is appropriate. In addition, it lacks tables that would be very useful and can contain critical information.
The article contains a lot of useful information on the issue. It is quite clear what is already known about this topic and the research question is clearly outlined. The abstract is too brief and Discussion section involve too much information. There must be a balance in the manuscript
Specifically
Introduction
Introduction section doesn’t involve too much information, while Conclusion section is too long. There is an asymmetry in the manuscript.
Research methods
Method is unclear. The authors must explain with more details.
Results
Results are unclear. The authors must explain with more details.
Tables with demographic and other information are necessary.
Positive: There are some strengths of the article that could have an impact in the field, such as the topic and its impact on the existed literature. The manuscript is approved after major changes.
Reviewer 3 Report
Thank you for the opportunity to read this paper. The authors propose A Systematic Review on Tri-response police, ambulance, mental health crisis models in 2 reducing involuntary detentions of mentally ill people.
The paper refers to an exciting and relevant research topic with significant theoretical and practical implications. Although it is an empty systematic review, I enjoyed reading the paper and I consider the authors were able to justify its results presenting a very clear, objective and methodologically accurate manuscript.
Although its strengths, I will present some small constructive comments and suggestions which may help the authors to improve their work.
- I consider the abstract and the 1. Background sections need further development and clarification.
- The background section doesn’t bring any added value and only confuses the reader regarding the difference between PACER and Tri-response models. The abstract should be more clear in terms of novelty, goal, main results and practical implications.
- In the beginning of the paper, it is not clear to me the relationship / difference between Police, Ambulance, Clinician, Early Response (PACER) mobile crisis service and similar tri-response models. This clarification should be present both in the abstract and in the background section.
- The reference list is not according to the journal guidelines.
I hope these comments and suggestions help the authors to improve the quality and readability of their manuscript.
Good luck!